# Automatic Pronunciation Assessment - A Review

**Yassine El Kheir, Ahmed Ali** and **Shammur Absar Chowdhury**
Qatar Computing Research Institute, HBKU, Qatar
shchowdhury@hbku.edu.qa

## Abstract

Pronunciation assessment and its application in computer-aided pronunciation training (CAPT) have seen impressive progress in recent years. With the rapid growth in language processing and deep learning over the past few years, there is a need for an updated review. In this paper, we review methods employed in pronunciation assessment for both phonemic and prosodic. We categorize the main challenges observed in prominent research trends, and highlight existing limitations, and available resources. This is followed by a discussion of the remaining challenges and possible directions for future work.

## 1 Introduction

Computer-aided Pronunciation Training (**CAPT**) technologies are pivotal in promoting self-directed language learning, offering constant, and tailored feedback for secondary language learners. The rising demand for foreign language learning, with the tide of globalization, fuels the increment in the development of CAPT systems. This surge has led to extensive research and development efforts in the field (Neri et al., 2008; Kang et al., 2018; Rogerson-Revell, 2021). CAPT systems have two main usages: (*i*) pronunciation assessment, where the system is concerned with the errors in the speech segment; (*ii*) pronunciation teaching, where the system is concerned with correcting and guiding the learner to fix mistakes in their pronunciation.

This paper addresses the former – focusing on pronunciation assessment, which aims to automatically score non-native speech-segment and give meaningful feedback. To build such a robust pronunciation assessment system, the following design aspects should be addressed.

**Modelling** Mispronunciation detection and diagnosis (MDD), in many cases, are more challenging to model compared to the vanilla automatic speech recognition (ASR) system, which converts speech into text regardless of pronunciation mistakes. Robust ASR should perform well with all variation including dialects and non-native speakers. However, MDD should mark phonetic variations from the learner, which may sometimes be subtle differences (Li et al., 2016a).

**Training Resources** Recent success in deep learning methods emphasized the need for in-domain training data. Language learners can be divided into two groups: adult secondary (L2) language learners and children language learners – the former depends on whether to build a system that is native language dependant (L1). At the same time, the latter identifies the need for children's voice, which is a challenging corpus to build (Council III et al., 2019; Venkatasubramaniam et al., 2023), even the accuracy for ASR for children is still behind compared to adult ASR (Liao et al., 2015). The scarcity and imbalanced distribution of negative mispronunciation classes pose a significant challenge in training data.

**Evaluation** There is no clear definition of right or wrong in pronunciation, instead an entire scale from unintelligible to native-sounding speech (Witt, 2012). Given that error in pronunciation is difficult to quantify, it can be split into (a) *Objective evaluations* – (*i*): phonetic or segmental; (*ii*): prosodic or supra-segmental; and (*iii*) place or articulation, manner of speech or sub-segmental; (b) *Subjective evaluations*; in many cases measured through listening tasks followed by human judgment, and can be split into three main classes: (*i*) intelligibility; (*ii*) comprehensibility and (*iii*) accentedness (or linguistic native-likeness). See Figure 1 for common pronunciation assessment factors.

Several studies have summarized advances in pronunciation error detection (Eskenazi, 1999, 2009; Witt, 2012; Li et al., 2016a; Chen and Li, 2016; Zhang et al., 2020; Caro Anzola and Mendoza Moreno, 2023). Eskenazi (1999) investigated

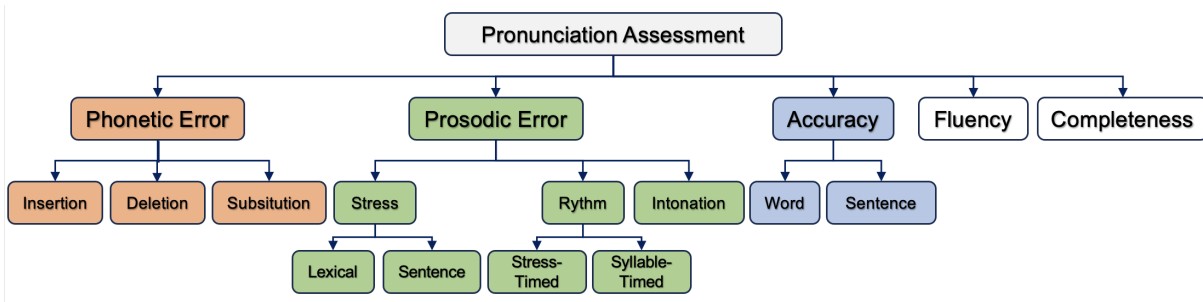

Figure 1: Types of Pronunciation Errors for Assessment

the potentials and limitations of ASR for L2 pronunciation assessment, showcasing its practical implementation using an interface developed at CMU. Furthermore, the study reports different automatic scoring techniques, emphasizing modalities of interaction, associated algorithms, and the challenges. Witt (2012) presented an overview of pronunciation error detection, encompassing various scoring methodologies and assessing commercial CAPT systems. Chen and Li (2016) provided a research summary, focusing on phoneme errors and prosodic error detection. More recently, Zhang et al. (2020) provided a summary of two automatic scoring approaches (a) ASR-based scoring to calculate confidence measures; and (b) acoustic phonetics scoring focusing on comparing or classifying phonetic segments using various acoustic features.

With large transformer-based pre-trained models gaining popularity, re-visiting the existing literature and presenting a comprehensive study of the field is timely. We provide an overview of techniques adapted for detecting mispronunciation in *(a)* segmental space, *(b)* assessing pronunciation with supra-segmal measures, along with *(c)* different data generation/augmentation approaches. Unlike previous overview studies, we also cover a handful of *(d)* qualitative studies bringing together the notion of intelligibility, comprehensiveness, and accentedness. We note the resources and evaluation measures available to the speech community and discuss the main challenges observed within prominent research trends, shedding light on existing limitations. Additionally, we also explore potential directions for future work.

## 2 Nuances of Pronunciation

Pronunciation can be defined as "the way in which a word or letter is said, or said correctly, or the way in which a language is spoken" .[1] Compared

---

[1] https://dictionary.cambridge.org/dictionary/english/pronunciation, Accessed: 2023-06-21

to other language skills, learning pronunciation is difficult. Yet, for learners, mastering L2 pronunciation is most crucial for better communication. Historically, pronunciation errors (mispronunciations) are characterized by phonetic (segmental) errors and prosodic (supra-segmental) errors (Witt, 2012; Chen and Li, 2016), as represented in Figure 1. This characterization provides some clear distinctions for pronunciation assessment.

### 2.1 Pronunciation Errors

**Phonetic Errors**

Phonetic (segmental) errors involve the production of individual sounds, such as vowels, and consonants, and it includes three errors: insertion, deletion, and substitution. This can be attributed to several factors, including negative language transfer, incorrect letter-to-sound conversion, and misreading of text prompts (Meng et al., 2007b; Qian et al., 2010; Kartushina and Frauenfelder, 2014; Li et al., 2016a). For example, Arabic L1 speakers may find it difficult to differentiate between /p/ and /b/ as the phoneme /p/ is non-existent in Arabic, so verbs like /park/ and /bark/ might sound similar to Arabic L1 speakers. Similarly, in Spanish, there are no short vowels, so words like /eat/ and /it/ might sound similar to Spanish L1 speakers.

**Prosodic Errors**

Prosodic features encompass elements that influence the pronunciation of an entire word or sentence, including stress, rhythm, and intonation. Errors related to prosodic features involve the production of larger sound units. For intelligibility, prosodic features particularly play a significant role (Raux and Kawahara, 2002). This is especially true for tonal languages (Dahmen et al., 2023) where variation in the pitch can lead to words with different meanings. Prosodic errors are often language-dependent and categorized by: stress (lexical and sentence), rhythm, and intonation.

| Corpus | Languages (L2) | Native Language (L1) | Dur/Utt | #Speakers | Reported SOTA Results / Relevant Studies |
|---|---|---|---|---|---|
| ISLE (Menzel et al., 2000) ∗ | English | German and Italian | 18/ | 46 | # PER:(Hosseini-Kivanani et al., 2021). Accent PCC: 68% (Rasipuram et al., 2015) |
| ERJ (Minematsu et al., 2004) ∗ | English | Japanese | /68,000 | 200 | # Utterance PCC (Luan et al., 2012). Word Intelligibility (Minematsu et al., 2011). Phoneme Errors (Ito et al., 2005) |
| CU-CHLOE (Meng et al., 2007a) | English | Cantonese and Mandarin | 34.6/18,139 | 210 | Phoneme F1-measure: 80.98% (Wu et al., 2021) |
| EURONOUNCE (Cylwik et al., 2009) | Polish | German | /721 | 18 | # Utterance rythm (Wagner, 2014) |
| iCALL (Chen et al., 2015) + | Mandarin | 24 countries | 142/90,841 | 305 | FAR: 8.65%, FRR: 3.09%: (Li et al., 2017). Tone Recognition: (Tong et al., 2015) |
| SingaKids-Mandarin (?) + | Mandarin | Singaporean (English) | 125/79,843 | 255 | PER: 28.51%. Tone Recognition (Tong et al., 2017) |
| SHEFCE (Ng et al., 2017b) ∗ | English, Cantonese | English, Cantonese | 25/ | 31 | Madarin syllabe error rate: 17.3%, English PER: 34.5% (Ng et al., 2017a) |
| VoisTUTOR (Yarra et al., 2019; Pal et al., 2022) | English | Kannada, Malayalam, Telugu, Tamil, Hindi and Gujarati | 14/26,529 | 16 | Word Intelligibility Accuracy: 96.58% (Anand et al., 2023) |
| EpaDB (Vidal et al., 2019a) + | English | Spanish | /3,200 | 50 | (Sancinetti et al., 2022) reported Min-Cost per phoneme |
| SELL-CORPUS (Chen et al., 2019) ∗ | English | Chinese | 31.6/ | 389 | F1-score Accent Detection: Word-level 35%, Sentence-level 45% (Kyriakopoulos et al., 2020) |
| L2-ARCTIC (Zhao et al., 2018a) ∗ | English | Hindi, Korean, Mandarin, Spanish, and Arabic | 3.6/ | 24 | F1-score: 63.04% (Lin and Wang, 2022a) |
| Speechocean762 (Zhang et al., 2021b) ∗ | English | Chinese | /5,000 | 250 | Phone PCC: 65.60% (Chao et al., 2022). Word Accuracy PCC: 59.80% (Chao et al., 2022). Word Stress PCC: 32.30% (Do et al., 2023). Sentence total score PCC: 79.60% (Chao et al., 2022) |
| LATIC (ZHANG, 2021) ∗ | Mandarin | Russian, Korean, French, and Arabic | 4/2,579 | 4 | Sentence Accuracy PCC: 69.80% (Lin and Wang, 2023b) |
| Arabic-CAPT (Algabri et al., 2022) | Arabic | India, Pakistan, Indonesia, Nepal, Afghanistan, Bangladesh, Nigeria, Uganda | 2.3/1,611 | 62 | F1-score 70.53% (Algabri et al., 2022) |
| AraVoiceL2 (EL Kheir et al., 2023b) | Arabic | Turkey, Nigeria, Bangladesh, Indonesia, Malaysia | 5.5/7,062 | 11 | F1-score 60.00% (EL Kheir et al., 2023b) |

Table 1: Widely used datasets. * represent publicly available dataset, + is available on request, # relevant study, Dur: total duration in hours, Utt: total number of utterances, SOTA: is the notable reported state-of-the-art for each corpus. FAR: false acceptance rate, FRR: false rejection rate, PCC: pearson correlation coefficient with human scores

*Stress* is the emphasis placed on certain syllables in a word or sentence. It is articulated by increasing the loudness, duration, and pitch of the stressed syllable. It can be categorized as *lexical stress*, if the stress is placed on syllables within the word, or *sentence stress* if the stress is placed on words within sentences. Mandarin learners of English have contrastive stress at the word-level that is absent in Korean, Mandarin speakers can have an advantage over Korean speakers in stress processing of English words (Wang, 2022).

*Rythm* is the pattern of stressed and unstressed syllables in a word or sentence. A language can be classified as either stress-timed or syllable-timed (Ohata, 2004; Matthews, 2014). In stress-timed languages, the duration of stressed syllables tends to dominate the overall time required to complete a sentence. Conversely, in syllable-timed languages, each syllable receives an equal amount of time during production.

*Intonation* refers to the melodic pattern and pitch variations in speech. L2 learners of Vietnamese and Mandarin Chinese encounter significant difficulty in acquiring distinct tones, particularly if their native language lacks tonality. Such tonal languages rely on different pitch patterns to convey distinct meanings, making it challenging for learners to accurately grasp and reproduce these tonal variations (Nguyen et al., 2014; Chen et al., 2015).

## 2.2 Pronunciation Constructs

The motivation behind mastering L2 pronunciation is to communicate properly in the target language. Most of the time, these successes are measured using three pronunciation constructs (Uchihara, 2022) – **Intelligibility**, **Comprehensibility**, and **Accentedness**. These are perceived measures, that are partially independent with overlapping features.

**Intelligibility** can be defined using the accuracy of the sound, word, and utterance itself along with utterance-level completeness (Abercrombie, 1949; Gooch et al., 2016). Accuracy refers where the learner pronounces each phoneme, or word in the utterance correctly. In contrast, completeness measures the percentage of words pronounced compared to the total number of words.

**Comprehensibility**, on the other hand, is defined based on the perceived ease or difficulty that listeners experience when understanding L2 speech. Fluency, defined by the smoothness of pronunciation and correct usage of pauses (Zhang et al., 2021b), is observed to be one of the key factors that determine the level of comprehensibility, along with good linguistic-knowledge and discourse-level organization (Trofimovich and Isaacs, 2012; Saito et al., 2016).

Among the three constructs, **accentedness**, which is defined as "listeners' perceptions of the degree to which L2 speech is influenced by their native language and/or colored by other non-native features" (Saito et al., 2016). It is often confused with both comprehensibility and intelligibility, influencing pronunciation assessment. The accent is an inherent trait that defines a person's identity and is one of the first things that a listener notices. It is often observed that most of the unintelligible speech is identified as highly accented whereas highly accented speech is not always unintelligible (Derwing and Munro, 1997; Kang et al., 2018; Munro and Derwing, 1995). Thus accents complicate fine-grained pronunciation assessment as it is harder to pinpoint (supra-)segment-level error.

## 3 Datasets

Obtaining datasets for pronunciation assessment is often challenging and expensive. Most of the available research work focused on private data, leaving only a handful of publicly accessible data to the research community. Table 1 provides an overview of available datasets, indicating English as a popular choice for the target language. Within this handful of datasets, a few datasets include phonetic/segmental-level transcription and even fewer provide manually rated word and sentence-level prosodic features, fluency along with overall proficiency scores offering insights to learner's L2 speech intelligibility and comprehensiveness (Arvaniti and Baltazani, 2000; ?; Cole et al., 2017; Zhang et al., 2021b). More details on datasets and annotation are in Appendix A and B respectively.

## 4 Research Avenues

In this section, we will delve into diverse approaches, old, revised, and current methodologies used for pronunciation modeling of both segmental and supra-segmental features, as illustrated in Figure 2 and Figure 3.

### 4.1 Classification based on Acoustic Phonetics

Classifier-based approaches explored both segmental and prosodic aspects of pronunciation. *Segmental* approaches involve the use of classifiers targeting specific phoneme pair errors, utilizing different acoustic features such as Mel-frequency cepstral coefficients (MFCCs) along with its first and second derivative, energy, zero-cross, and spectral features (Van Doremalen et al., 2009; Huang et al., 2020), with different techniques such as Linear Discriminant Analysis (LDA) (Truong et al., 2004; Strik et al., 2009), decision trees (Strik et al., 2009). *Prosodic* approaches focus on detecting lexical stress and tones, utilizing features such as energy, pitch, duration, and spectral characteristics, with classifiers like Gaussian mixture models (GMMs) (Ferrer et al., 2015), support vector machines (SVMs) (Chen and Wang, 2010; Shahin et al., 2016), and deep neural network (DNNs) (Shahin et al., 2016), and multi-distribution DNNs (Li et al., 2018a).

### 4.2 Extended Recognition Network (ERN)

ERNs are neural networks used in automatic speech recognition to capture broader contextual information, they leverage enhanced lexicons in combination with ASR systems. They cover canonical tran-

scriptions as well as error patterns, enabling the detection of mispronunciations beyond standard transcriptions (Meng et al., 2007b; Ronen et al., 1997; Qian et al., 2010; Li et al., 2016b). However, ERNs often depend on experts or hand-crafted error patterns, which are typically derived from non-native speech transcriptions as illustrated in (Lo et al., 2010) which makes it language-dependent approach and may limit their generalizability when dealing with unknown languages.

### 4.3 Likelihood-based Scoring and GOP

The initial likelihood-based MD algorithms aim to detect errors at the phoneme level using pre-trained HMM-GMM ASR models. Notably, Kim et al. (1997) introduced a set of three HMM-based scores, including likelihood scores, log posterior scores, and segment-duration-based scores. Among these three, the log-based posterior scores are widely adopted due to their high correlation with human scores, and are also used to calculate the popular 'goodness of pronunciation' (GOP) measure. The GMM-HMM based GOP scores can be defined by the Equation 1.

$$GOP(p) = P(p|O) = \frac{p(O|p)\ P(p)}{\sum_q p(O|q)\ P(q)} \qquad (1)$$

$O$ denotes a sequence of acoustic features, $p$ stands for the target phone, and $Q$ represents the set of phones.
These scores are further improved using forced alignment framework (Kawai and Hirose, 1998). More details are presented in Witt and Young (2000).

### 4.4 Reformulations of GOP

To enhance further the effectiveness of GOP scoring, (Zhang et al., 2008) are first to propose a log-posterior normalized GOP defined as:

$$GOP_r(p) = |\frac{p(o_t|p)P(p)}{\max_q p(o_t|q)}| \qquad (2)$$

Building upon this, Wang and Lee (2012) adopted the GOP formulation and incorporate error pattern detectors for phoneme mispronunciation diagnosis tasks. With the emergence of DNN in the field of ASR, Hu et al. (2013, 2015a,b) demonstrated that using a DNN-HMM ASR for GOP yields improved correlation scores surpassing GMM-HMM based GOP. The GOP and its reformulation represent a significant milestone. It leverages pre-trained acoustic models on the target language without the

necessitating of speaker's L1 knowledge. Furthermore, it offers the advantage of being computationally efficient to calculate. However, these scores lack context-aware information that is crucial for accurate pronunciation analysis. To overcome this, Sudhakara et al. (2019) presented a context-aware GOP formulation by adding phoneme state transition probabilities (STP) extracted from HMM model to the GOP score calculation. Furthermore, Shi et al. (2020) proposed a context-dependent GOP, incorporating a phoneme duration factor $\alpha_i$, and phonemes transition factor $\tau$. The formulated GOP score combines all the contextual scores as illustrated in Equation 3.

$$E_t = -\sum p(q|O)\ log(p(q|O))$$
$$\tau(p) = \sum_t \frac{\frac{1}{E_t}}{\sum_{t'} \frac{1}{E_{t'}}} log(p(q|O)) \qquad (3)$$
$$GOP(p) = (1 - \alpha_i) * \tau(p)$$

For *sentence accuracy* evaluation, one common approach is to calculate the average GOP scores across phonemes (Kim et al., 1997; Sudhakara et al., 2019). However, relying solely on averaging GOP scores at the phoneme level is limited. A recent approach in (Sheoran et al., 2023) proposed a combination of phone feature score and audio pitch comparison using dynamic time warping (DTW) with an ideal pronounced speech, as a score to assess prosodic, fluency, completeness, and accuracy at the sentence level. Inspired by GOP, Tong et al. (2015) proposed Goodness of Tone (GOT) based on posterior probabilities of tonal phones.

While efforts have been made to improve the GOP formulation, it is important to acknowledge that the GOP score still has limitations, specifically in its ability to identify specific types of mispronunciation errors (deletion, insertion, or substitution), and it also demonstrates a degree of dependency on the language of the acoustic model.

### 4.5 End-to-End Modeling

In the new era of DNNs and Transformers, there is a significant exploration by researchers in leveraging the power of these models and training end-to-end pronunciation systems. Li et al. (2017) introduced LSTM mispronunciation detector leveraging phone-level posteriors, time boundary information, and posterior extracted from trained DNNs models on the classification of phonetic attributes (place, manner, aspiration, and voicing). In contrast, Kyriakopoulos et al. (2018) introduced

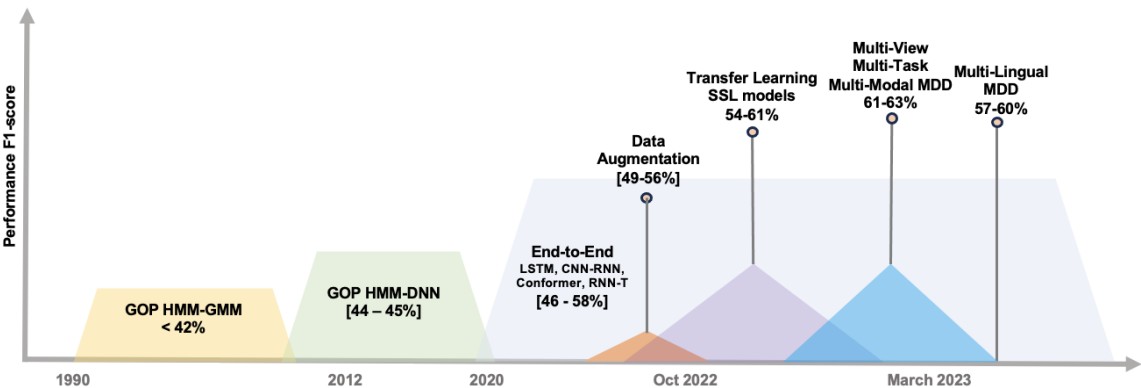

Figure 2: Overview of the performance of different phonetic pronunciation detection models on L2-ARCTIC

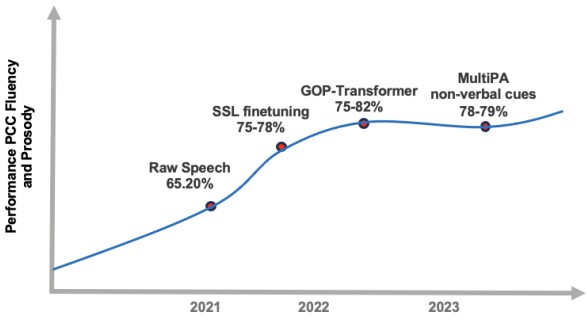

Figure 3: Overview of the performance of fluency and prosody assessment models on Speechocean762

a siamese network with BiLSTM for pronunciation scoring by extracting distance metrics between phone instances from audio frames. A notable approach presented in Leung et al. (2019) introduced a CNN-RNN-CTC model for phoneme mispronunciation detection without any alignment component dependency. Subsequently, Feng et al. (2020) incorporated character embeddings to enhance CNN-RNN model. Furthermore, Ye et al. (2022) enhanced the later model using a triplet of features consisting of acoustic, phonetic, and linguistic embeddings. Subsequently, GOP features extracted from pre-trained ASR are enhanced using a Transformer encoder to predict a range of scores of prosodic and segmental scores (Gong et al., 2022), or using additional SSL representation features, energy, and duration within the same architecture (Chao et al., 2022), or using Conformer encoder explored in (Fan et al., 2023). Moreover, PEPPANET is also a transformer-based mispronunciation model, but can jointly model the dictation process and the alignment process, and it provides corresponding diagnostic feedback (Yan

et al., 2023a). A subsequent improvement of PEP-PANET uses knowledge about phone-level articulation traits with a graph convolutional network (GCN) to obtain more discriminative phonetic embeddings (Yan et al., 2023b). Recently, Zhang et al. (2023) proposed recurrent neural network transducer RNN-T for L2 phoneme sequence prediction along with an extended phoneme set and weakly supervised training strategy to differentiate similar-sounding phonemes from different languages.

Several approaches have also been proposed for *supra-segmental features scoring*. Yu et al. (2015), proposed a new approach where traditional time-aggregated features are replaced with time-sequence features, such as pitch, to preserve more information without requiring manual feature engineering, a BiLSTM model is proposed for fluency predictions. Tao et al. (2016); Chen et al. (2018), studied different DNNs models such as CNN, BiLSTM, Attention BiLSTM to predict the fluency and prosodic scoring. (Lin and Wang, 2021) utilized deep features directly from the acoustic model instead of relying on complex feature computations like GOP scores with a scoring module, incorporating a self-attention mechanism, which is designed to model human sentence scoring. More recently, (Zhu et al., 2023) proposed BiLSTM model trained to predict the intelligibility score of a given phoneme or word segment using an annotated intelligibility L2 speech using shadowing.

Towards *lexical stress* detection, several methods have been proposed to improve accuracy and performance. Ruan et al. (2019) proposed a sequence-to-sequence approach using the Transformer model upon the need for long-distance contextual information to predict phoneme sequence with stress marks. Furthermore, Korzekwa et al. (2020a) intro-

duced an attention-based neural network focusing on the automatic extraction of syllable-level features that significantly improves the detection of lexical stress errors.

*Tone classification* has received significant attention in Mandarin language learning due to the crucial role that tones play in Mandarin Chinese. To address the challenge several methods have been proposed. One approach involves training a DNN to classify speech frames into six tone classes (Ryant et al., 2014). Inspired by this, DNNs have been used to map combined cepstral and tonal features to frame-level tone posteriors. These tone posteriors are then fed into tone verifiers to assess the correctness of tone pronunciation (Lin et al., 2018; Li et al., 2018b). Another study utilizes CNN to classify syllables into four Mandarin tones (Chen et al., 2016a). Similarly, ToneNet, a CNN-based network is introduced for Chinese syllable tone classification using mel-spectrogram as a feature representation (Gao et al., 2019). Additionally, a BiLSTM model is proposed as an alternative to capture long-term dependencies in acoustic and prosodic features for tone classification (Li et al., 2019).

### 4.6 Self-Supervised Models

Motivated by the recent success of self-supervised learning methods (Baevski et al., 2020; Hsu et al., 2021; Chen et al., 2022; Mohamed et al., 2022) in speech recognition and related downstream tasks such as emotion recognition, speaker verification, and language identification (Chen and Rudnicky, 2023; Fan et al., 2020), self-supervised approaches is employed also in this field.

Xu et al. (2021) explored finetuning wav2vec 2.0 on frame-level L2 phoneme prediction. A pretrained HMM-DNN ASR is used to extract time force-alignment. To overcome the dependency on time alignment, Peng et al. (2021) propose a CTC-based wav2vec 2.0 to predict L2 phonemes sequences. Building upon this work, Yang et al. (2022) propose an approach that leverages unlabeled L2 speech using momentum pseudo-labeling. In a contrasting approach, (Lin and Wang, 2022b) combined wav2vec 2.0 features and phoneme text embeddings in a jointly learning framework to predict frame-level phoneme sequence and detect boundaries. Recently, EL Kheir et al. (2023a) explored the multi-view representation utilizing mono- and multilingual wav2vec 2.0 encoders to capture different aspects of speech production and

leveraging articulatory features as auxiliary tasks to phoneme sequence prediction. Furthermore, Kheir et al. (2023b) introduces a novel L1-aware multilingual, L1-MultiMDD, architecture for addressing mispronunciation in multilingual settings encompassing Arabic, English, and Mandarin using wav2vec-large pre-trained model as the acoustic encoder. L1-MultiMDD is enriched with L1-aware speech representation, allowing it to understand the nuances of each speaker's native language.

SSL models have proven to be effective in predicting fluency and prosodic scores assigned by human annotators. Kim et al. (2022); Lin and Wang (2023a); Yang et al. (2022) fine-tuned wav2vec 2.0 and Hubert to predict prosodic and fluency scores. Similarly, another research conducted in (Lin and Wang, 2023a) jointly predicts L2 phoneme sequence using CTC loss, and predicts prosodic scores using fused acoustic representations with phoneme embeddings. Subsequently Lin and Wang (2023b) introduced a fusion of language embedding, representation features and build a unified framework for multi-lingual prosodic scoring. Recently, Chao et al. (2022); Kheir et al. (2023a); Chen et al. (2023), enriched latent speech extracted from SSL models with handcrafted frame- and utterance-level non-verbal paralinguistic cues such as duration, and energy for modeling Fluency and Prosody scores.

### 4.7 Unsupervised Approaches

It is important to note that the aforementioned approaches for studying mispronunciation detection typically involve the need for expert knowledge, laborious manual labeling, or dependable ASR results, all of which come with significant costs. In contrast, recent years have witnessed considerable endeavors in unsupervised acoustic pattern discovery, yielding sub-optimal outcomes. Lee and Glass (2012) initially investigated a comparison-based approach that analyzes the extent of misalignment between a student's speech and a teacher's speech. In subsequent studies Lee and Glass (2015); Lee et al. (2016), explored the discovery of mispronunciation errors by analyzing the acoustic similarities across individual learners' utterances, with a proposed n-best filtering method to resolve ambiguous error candidate hypotheses derived from acoustic similarity clustering. Furthermore, Mao et al. (2018) proposed *k*-means clustering on phoneme-based phonemic posterior-grams (PPGs) to expand the phoneme set in L2 speech. More recently, Sini et al.

(2023) introduced a weighted DTW alignment as an alternative to the GOP algorithm for predicting probabilities and the sequence of target phonemes. Their proposed method achieves comparable results to the GOP scoring algorithm, likewise Anand et al. (2023) explored alignment distance between wav2vec 2.0 representations of teacher and learner speech using DTW, to distinguish between intelligible and unintelligible speech.

## 4.8 Data Augmentation

Two major challenges in this field are L2 data scarcity and the imbalanced distribution of negative classes (mispronunciation). To address these challenges, researchers have opted for data augmentation techniques that are proven to be quite effective in pronunciation assessment. Such methods employed strategies like altering the canonical text by introducing mismatched phoneme pairs while preserving the original word-level speech (Fu et al., 2021). Additionally, a mixup technique is utilized in the feature space, leveraging phone-level GOP pooling to construct word-level training data (Fu et al., 2022). Furthermore, the error distance of the clustered SSL model embeddings are employed to substitute the phoneme sound with a similar sound (Zhang et al., 2022b). These latter approaches depend on the reuse of existing information rather than generating novel instances of mispronunciations. In (Fernandez et al., 2017), voice transformations in pitch, vocal-tract, vocal-source characteristics to generate new samples. Furthermore, L2-GEN can synthesize realistic L2 phoneme sequences by building a novel Seq2Seq phoneme paraphrasing model (Zhang et al., 2022a). Korzekwa et al. (2020b) proposed an augmentation technique by generating incorrectly stressed words using Neural TTS. Furthermore, Korzekwa et al. (2022) provided an overview of mispronunciation error generation using three methods, phoneme-2-phoneme P2P relies on perturbing phonetic transcription for the corresponding speech audio, text-2-speech create speech signals that match the synthetic mispronunciations, and speech-2-speech S2S to simulate a different aspect of prosodic nature of speech. Recently, SpeechBlender (EL Kheir et al., 2023b) framework is introduced as a fine-grained data augmentation pipeline that linearly interpolates raw good speech pronunciations to generate mispronunciations at the phoneme level.

## 5   Evaluation Metrics

**Phoneme Error Rate (PER):** is a common metric used in the MD evaluation, measuring the accuracy of the predicted phoneme with the human-annotated sequence. However, PER might not provide a comprehensive assessment of model performance when mispronunciations are infrequent which is the case for MD datasets.

**Hierarchical Evaluation Structure:** The hierarchical evaluation structure developed in (Qian et al., 2010), has also been widely adopted in (Wang and Lee, 2015; Li et al., 2016a; EL Kheir et al., 2023a) among others. The hierarchical mispronunciation detection depends on detecting the misalignment over: *what is said* (annotated verbatim sequence); *what is predicted* (model output) along with *what should have been said* (text-dependent reference sequence). Based on the aforementioned sequences, the false rejection rate, false acceptance rate, and diagnostic error rate are calculated, using:

- True acceptance (**TA**): the number of phones annotated and recognized as correct pronunciations.

- True rejection (**TR**): the number of phones both annotated and correctly predicted as mispronunciations. The labels are further utilized to measure the diagnostic errors and correct diagnosis based on the prediction output and text-dependent canonical pronunciation.

- False rejection (**FR**): the number of phones wrongly predicted as mispronunciations.

- False acceptance (**FA**): the number of phones misclassified as correct pronunciations.

As a result, we can calculate the false rejection rate (**FRR**) that indicates the number of phones recognized as mispronunciations when the actual pronunciations are correct, false acceptance rate (**FAR**) that indicates phones misclassified as correct but are actually mispronounced, and diagnostic error rate (**DER**) using the following equations:

$$FRR = \frac{FR}{TA + FR} \quad (4)$$

$$FAR = \frac{FA}{FA + TR} \quad (5)$$

$$DER = \frac{DE}{CD + DE} \quad (6)$$

**Precision, Recall, and F-measure** are also widely used as the performance measures for mispronunciation detection. These metrics are defined as follows:

$$Precision = \frac{TR}{TR + FR} \qquad (7)$$

$$Recall = \frac{TR}{TR + FA} = 1 - FAR \qquad (8)$$

$$F - measure = 2 \cdot \frac{Precision \cdot Recall}{Precision + Recall} \qquad (9)$$

**Pearson Correlation Coefficient:** PCC is widely used to measure the relation of the predicted score of fluency, stress, and prosody with other supra-segmental and pronunciation constructs with subjective human evaluation for pronunciation assessment. The human scores are typically averaged across all annotators to provide a comprehensive score.

# 6 Challenges and Future Look

There are two significant challenges facing advancing the research further: (1) the lack of public resources. Table 1 shows a handful of L2 languages. With 7,000 languages spoken on earth, there is an urgent need for inclusivity in pronouncing the languages of the world. (2) There is a need for a unified evaluation metric for pronunciation learning, this can be used to establish and continually maintain a detailed leaderboard system, which serves as a dynamic and multifaceted platform for tracking, ranking, and showcasing the advances in the field to guide researchers from academia and industry to push the boundaries for pronunciation assessments from unintelligible audio to native-like speech. The advent of AI technology represents a pivotal moment in our technological landscape, offering the prospect of far-reaching and transformative changes that have the potential to revolutionize a wide array of services in CAPT. Listing here some of the opportunities:

**Integration with Conversation AI Systems:** The progress made in Generative Pre-trained Transformer (GPT) led to a human-like text-based conversational AI. Furthermore, low-latency ASR has enhanced the adoption of speech processing in our daily life. Both have paved the way for the development of a reliable virtual tutor CAPT system, which is capable of interacting and providing students with instant and tailored feedback, thereby enhancing their pronunciation skills and augmenting private tutors.

**Multilingual:** Recent advancements in end-to-end ASR enabled the development of multi-lingual code-switching systems (Datta et al., 2020; Chowdhury et al., 2021; Ogunremi et al., 2023). The great progress in SSL expanded ASR capabilities to support from over 100 (Pratap et al., 2023), to over 1,000 (Pratap et al., 2023) languages. Traditional research in pronunciation assessments focused on designing monolingual assessment systems. However, recent advancements in multilingualism allowed for the generalization of findings across different languages. Zhang et al. (2021a) explored the adaptation of pronunciation assessments from English (a stress-timed language) to Malay (a syllable-timed language). Meanwhile, Lin and Wang (2023b) investigated the use of language-specific embeddings for diverse languages, while optimizing the entire network within a unified framework.

**Children CAPT:** There is a noticeable imbalance in research between children learning pronunciation research papers, for example, reading assessments, compared to adults' L2 language learning. This disparity can be attributed to the scarcity of publicly available corpora and the difficulties in collecting children's speech data.

**Dialectal CAPT:** One implicit assumption in most of the current research in pronunciation assessment is that L2 is a language with a standard orthographic rule. However, Cases like dialectal Arabic – which is every Arab native language, there is no standard orthography. Since speaking as a native is the ultimate objective for advanced pronunciation learning, there is a growing demand for this task.

# 7 Conclusion

This paper serves as a comprehensive resource that summarizes the current research landscape in automatic pronunciation assessment covering both segmental and supra-segmental space. The paper offers insights into the following:

- Modeling techniques – highlighting design choices and their effect on performance.
- Data challenges and available resources – emphasizes the success of automatic data generation/augmentation pipeline and lack of consensus annotation guidelines and labels. The paper also lists available resources to the community along with the current state-of-the-art performances reported per resource.
- Importance of standardised evaluation metrics and steady benchmarking efforts.

With the current trend of end-to-end modeling and multilingualism, we believe this study will provide a guideline for new researchers and a foundation for future advancements in the field.

## Limitations

In this overview, we address different constructs of pronunciation and various scientific approaches for detecting errors, predicting prosodic and fluency scores among others. However, we have not included the corrective feedback mechanism of CAPT system. Moreover, the paper does not cover, in detail, the limited literature on CAPT's user study, or other qualitative study involving subjective evaluation. With the fast-growing field of pronunciation assessments, it is hard to mention all the studies and resources. Therefore, we would also like to apologize for any oversights of corpora or major research papers in this study.

## Ethics Statement

We discussed publicly available research and datasets in our study. Any biases are unintended.

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

# Appendix

## A   Mispronunciation Assessment and Non-Native Datasets

In this section, we provide a comprehensive overview of existing pronunciation assessment datasets as presented in Table 1.

### A.1   ISLE Speech Corpus (Menzel et al., 2000)

The ISLE corpus stands out as one of the largest speech corpora in terms of duration and offers the advantage of being distributed by ELDA. This corpus focuses on German and Italian accented English, featuring recordings of 23 intermediate-level speakers from each accent group. The participants, primarily employees and students from project sites in Italy, Germany, and the UK, were selected to achieve a balance of native languages (German/Italian) while including a small number of non-native speakers from other countries (Spanish, French, Chinese) and native British English speakers for comparison purposes. The corpus contains readings of both nonfictional autobiographic text (1300 words) and short utterances (1100 words) designed to cover common pronunciation errors made by language learners. It offers annotations at both the word and phone levels, making it particularly valuable for developing Computer Assisted Language Learning systems. The annotation process involved multiple steps, including quality checks, reference transcription, forced alignment, and the addition of canonical pronunciations and stress markings. An emphasis was placed on matching non-English phones to the closest equivalent in the UK English phone set, with occasional input from a trained phonetician and a native speaker of the speaker's mother tongue for verification and quality improvement purposes.

### A.2   English read by Japanese Corpus (ERJ) (Minematsu et al., 2004)

The ERJ corpus (English Read by Japanese) is a database of English speech read by Japanese students. It was created to support research in computer-assisted language learning (CALL). The corpus contains 800 utterances from 202 (100 males and 102 females) Japanese university students, each of whom read a set of 100 sentences. The sentences were selected to be phonetically balanced and to cover a variety of grammatical structures. The corpus is annotated with phonemic transcriptions and prosodic markings. The ERJ corpus consists of two sets of data: a phonemic pronunciation set and a prosody set. The phonemic pronunciation set contains 460 phonetically-balanced sentences, 32 sentences including phoneme sequences difficult for Japanese to pronounce correctly, and 100 sentences designed for the test set. The prosody set contains 94 sentences with various intonation patterns, 120 sentences with various accent and rhythm patterns, and 109 words with various accent patterns. The ERJ corpus is annotated with phonemic transcriptions and prosodic markings. The phonemic transcriptions are based on the International Phonetic Alphabet (IPA). The prosodic markings include information about intonation, accent, and rhythm.

### A.3   Chinese University Chinese Learners of English (CU-CHLOE) (Meng et al., 2007a)

The CU-CHLOE corpus encompasses a diverse group of speakers, including 110 Mandarin speakers (60 males and 50 females) and 100 Cantonese speakers (50 males and 50 females). It is structured into five distinct sections, namely confusable words, minimal pairs, phonemic sentences, the Aesop's Fable "The North Wind and the Sun," and prompts sourced from the TIMIT dataset. Trained linguists have diligently labeled all sections, except for the TIMIT prompts, contributing to approximately 30% of the comprehensive CHLOE data. This expert annotation ensures the corpus provides reliable and precise linguistic information, making it a valuable resource for exploring Mandarin and Cantonese speech characteristics and facilitating advancements in research within these languages.

### A.4   EURONOUNCE (Cylwik et al., 2009)

EURONOUNCE is a speech corpus developed for an ASR-based pronunciation tutoring system. The annotation conveys a phonetic segmentation, with identifying pronunciation errors, including substitutions, insertions, and deletions. The resulting annotations are then reviewed by a native speaker of the source language (German) to validate the assessment.

### A.5   iCALL (Chen et al., 2015)

iCALL is comprised of 90,841 spoken statements delivered by 305 individuals, spanning a cumulative duration of 142 hours. The speaker composition ensures gender equality, encompasses various native languages, and reflects a representative range of adult Mandarin learners. These oral statements

have been transcribed phonetically and evaluated for fluency by proficient native Mandarin speakers.

### A.6 SingaKids-Mandarin (Chen et al., 2016b)

SingaKids-Mandarin is a comprehensive speech corpus consisting of recordings from 255 Singaporean children between the ages of 7 and 12. The corpus aims to provide a resource for studying Mandarin Chinese pronunciation and language acquisition in Singaporean children. The corpus contains a total of 125 hours of audio data, with 75 hours dedicated to speech. Within this dataset, there are 79,843 utterances, each of which has been meticulously annotated by human experts. The annotations include phonetic transcriptions, lexical tone markings, and proficiency scoring at the level of individual utterances. The reading scripts used in the corpus encompass a wide range of utterance styles. They cover syllable-level minimal pairs, individual words, phrases, complete sentences, and even short stories. This diversity allows for a thorough analysis of different aspects of Mandarin pronunciation and fluency in Singaporean children.

### A.7 SHEFCE (Ng et al., 2017b)

SHEFCE (ShefCE) is a bilingual parallel speech corpus that focuses on Cantonese and English. It was recorded by second language (L2) English learners in Hong Kong. The corpus consists of recordings from 31 undergraduate to postgraduate students, aged 20 to 30. The corpus includes a total of 25 hours of speech data, with approximately 12 hours recorded in Cantonese and 13 hours recorded in English. The primary goal of this corpus is to provide a resource for studying the speech patterns, pronunciation, and language acquisition of Cantonese-speaking individuals who are learning English as a second language.

### A.8 L2-ARCTIC (Zhao et al., 2018a)

The L2-ARCTIC[2] corpus is a specialized speech corpus designed for research in voice conversion, accent conversion, and mispronunciation detection in non-native English. It encompasses a substantial collection of 26867 utterances from 24 non-native speakers (12 males and 12 females) whose $L1$ languages include Hindi, Korean, Mandarin, Spanish, Arabic, and Vietnamese. The recordings were sourced from a total of 4 speakers per $L1$ language, consisting of 2 males and 2 females ensuring a balanced distribution in terms of gender and native

---

[2]version 5 released in 2020 avalaible: https://psi.engr.tamu.edu/l2-arctic-corpus

languages (L1s). Yet, only 150 utterances is manually per speaker to identify three types of segmental mispronunciation errors: substitutions, deletions, and insertions resulting in 3.66 hours.

### A.9 VoisTUTOR corpus (Yarra et al., 2019)

VoisTUTOR is a pronunciation assessment corpus of Indian second language (L2) learners learning English. The corpus consists of audio recordings of 16 Indian L2 learners reading a set of 1676 sentences. The recordings are accompanied by phonetic transcriptions, human ratings of pronunciation accuracy on a scale of 0 to 10 for each utterance, and binary decisions for seven factors that affect pronunciation quality: intelligibility, phoneme quality, phoneme mispronunciation, syllable stress quality, intonation quality, correctness of pauses, and mother tongue influence.

### A.10 SELL-CORPUS (Chen et al., 2019)

SELL-CORPUS is a multiple accented speech corpus for L2 English learning in China. The corpus consists of audio recordings of 389 volunteer speakers, including 186 males and 203 females. The speakers are from seven major regional dialects of China, including Mandarin, Cantonese, Wu, Min, Hakka, and Southwestern Mandarin. The corpus contains 31.6 hours of speech recordings. Each recording in the corpus contains a word-level orthographic transcription manually inspected and cleaned by inserting, substituting, or deleting mismatching characters.

### A.11 English Pronunciation by Argentinians Database (EpaDB) (Vidal et al., 2019a)

EpaDB consists of English phrases recorded by native Spanish speakers with varying levels of English proficiency. The recordings are annotated with ratings indicating the quality of pronunciation at the phrase level. Additionally, detailed phonetic alignments and transcriptions are provided, indicating which phones were actually pronounced by the speakers.

### A.12 Speechocean762 (Zhang et al., 2021b)

Speechocean762 is an extensive dataset specifically designed for pronunciation assessment. It comprises a total of 5,000 English utterances obtained from 250 non-native speakers. Each utterance in the dataset is associated with five aspect scores at the utterance level, namely accuracy, fluency, completeness, prosody, and a total score ranging from 0 to 10. Additionally, for each word within the ut-

terance, three aspect scores are provided, including accuracy, stress, and a total score ranging from 0 to 10. Furthermore, an accuracy score is assigned to each individual phoneme, ranging from 0 to 2. To ensure reliability, each of these scores is annotated by five expert evaluators.

### A.13 LATIC (ZHANG, 2021)

LATIC primarily targets non-native learners of Mandarin Chinese. The dataset comprises four hours of recordings involving specifically selected non-native Chinese speakers. The participants' L1's vary, including Russian, Korean, French, and Arabic. Following each audio file, annotators transcribed the "closest" transcript and provided modern Mandarin annotations after careful listening.

### A.14 Arabic-CAPT (Algabri et al., 2022)

Arabic-CAPT is an Arabic mispronunciation detection corpus consisting of 62 non-native Arabic speakers from 20 different nationalities, totaling 2.36 hours of speech data. The Arabic non-native speech is annotated following the guidelines in (Zhao et al., 2018a).

### A.15 AraVoiceL2 (EL Kheir et al., 2023b)

AraVoiceL2 is an Arabic mispronunciation detection corpus comprised of 5.5 hours of data recorded by 11 non-native Arabic speakers. Each speaker recorded a fixed list of 642 words and short sentences, making for a total of 7, 062 recordings. The corpus is annotated at character level including diacritics following (Zhang et al., 2021b) guidelines.

### A.16 Non-Native Datasets:

Table 2 provides a comprehensive overview of existing non-native datasets that are particularly beneficial as they enable the extraction of error patterns allowing for a thorough assessment of L2 pronunciation. These datasets can also be used to train robust ASR models, from which we can extract valuation features to accurately score L2 speech. Furthermore, non-native datasets can enhance existing pronunciation assessment end-to-end approaches.

## B Annotation

In this section, we provide an overview of the standard approaches to annotate segmental and supra-segmental errors widely used in MDD research.

### B.1 Segmental Annotation

Segmental human annotation can be approached from two perspectives. The first and the commonly utilized approach in most available MDD corpora involves linguistics experts transcribing the actual sequence of phonemes spoken by the learner (Bonaventura et al., 2000; Zhao et al., 2018b; Vidal et al., 2019b). The resulted transcription is commonly referred as hypothesis annotation. Additionally, extra tasks can be incorporated, such as providing time boundaries for each pronounced phoneme, to further enhance the annotation process. This approaches may have limitations in capturing non-clear speech instances, such as heavily accented pronunciations that may not be easily detected by human annotators. This leads to the second approach, which incorporates scoring-based methods in addition to hypothesis annotation (Zhang et al., 2021c). In this approach, a score is assigned to each phoneme: 0 represents deleted or mispronounced phonemes, 1 indicates heavily accented pronunciation, and 2 signifies good pronunciation. This scoring-based approach provides a more comprehensive assessment of pronunciation quality, particularly in cases where clear detection by human annotators may be challenging.

### B.2 Supra-segmental Annotation

Limited research has been conducted regarding the annotation of supra-segmental features at the rhythm, stress, and intonation levels such as in (Arvaniti and Baltazani, 2000; Chen et al., 2016b; Cole et al., 2017). However, the ultimate objective of annotating these supra-segmental aspects is to ensure the fluency and intelligibility of L2 learners' speech. Hence the most commonly used annotated datasets at the prosodic level provide human-scored words, and sentences based on overall pronunciation quality and fluency. Multiple tiers of human scoring annotations can be applied in this context. This includes providing the accuracy of pronounced words to assess their intelligibility, assigning scores to evaluate the positioning of stress within individual words or within sentences, and evaluating sentence fluency by considering factors such as pauses, repetitions, and stammering in speech as adapted in (Zhang et al., 2021c).

| Corpus | Languages | Dur / #Utt | #Speakers |
|---|---|---|---|
| Demuth Sesotho Corpus (Demuth, 1992) | Sesotho | 98h / / 13250 | 4 |
| TIDIGITS (Leonard and Doddington, 1993) | English | - | / 326 |
| CMU Kids Corpus (Eskenazi et al., 1997) | English | / 5180 | 76 |
| CU Children's Read and Prompted Speech Corpus (Hagen et al., 2003) | English | / 100 | 663 |
| CU Story Corpus (Hagen et al., 2003) | English | 40h / 7062 | 106 |
| PF-STAR Children's Speech Corpus (Batliner et al., 2005) | English | 14.5h / | 158 |
| TBALL (Kazemzadeh et al., 2005) | English | 40h / 5000 | 256 |
| Swedish NICE Corpus (Bell et al., 2005) | Swedish | - | 5580 |
| Providence Corpus (Demuth et al., 2006) | English | 363h / | 6 |
| Lyon Corpus (Demuth and Tremblay, 2007) | French | 185h / | 4 |
| CHIEDE (Garrote, 2008) | Spanish | 8h / / 15444 | 59 |
| CFSC (Pascual and Guevara, 2012) | Filipino | 8h / | 57 |
| CASSCHILD (Gao et al., 2012) | Mandarin | - | 23 |
| CALL-SLT (Rayner et al., 2014) | German | / 5000 | - |
| Boulder Learning—MyST Corpus (Boulder Learning Inc, 2019) | English | 393h / 228874 | 1371 |
| TLT-school (Gretter et al., 2020) | English and German | 119.1h / 26059 | 6547 |

Table 2: Non-Native Speech Datasets