# OpenReview forum: "Automatic Pronunciation Assessment - A Review"
_EMNLP/2023/Conference — EMNLP 2023 Findings_

### Official Review · Reviewer_hbGh · 2023-07-27

**Soundness:** 4

**Excitement:**

3: Ambivalent: It has merits (e.g., it reports state-of-the-art results, the idea is nice), but there are key weaknesses (e.g., it describes incremental work), and it can significantly benefit from another round of revision. However, I won't object to accepting it if my co-reviewers champion it.

**Paper Topic And Main Contributions:**

The paper provides a review of work on pronunciation assessment over the last two decades.  It discusses the types of error that can occur, training and evaluation datasets, and the historical development of assessment methods spanning HMM based likelihood measures to modern self-supervised pre-trained end-to-end models.  It is very high level and essentially a literature review.


**Reasons To Accept:**

The paper is clearly written and provides a focussed but reasonably comprehensive survey of published work on pronunciation assessment.  It will provide a valuable entry point to anybody entering the field of pronunciation assessment.  The information on the available datasets is also useful.


**Reasons To Reject:**

There is nothing original in the paper but equally there is nothing to object to.


**Reproducibility:**

N/A: Doesn't apply, since the paper does not include empirical results.

**Reviewer Confidence:**

4: Quite sure. I tried to check the important points carefully. It's unlikely, though conceivable, that I missed something that should affect my ratings.

---

> ### Author Rebuttal · Authors · 2023-08-29
>
> We thank the reviewer for the  constructive and positive feedback concerning our paper. Your recognition of its strengths is truly valued.
>
> We would like to highlight the distinctive contributions that our paper brings to the existing body of literature: Within this paper:
> (i) We offered a comprehensive exploration of pronunciation assessment, encompassing both phonetic and prosodic errors. (ii) Our approach fills a critical void left by previous methodologies that predominantly concentrated solely on the phonetic aspect of pronunciation assessment. (iii) Furthermore, we presented an extensive overview of previous, contemporary, and emerging methodologies in the field of pronunciation assessment, spanning the years 2013 to 2023. (iv) We also meticulously outlined the existing datasets employed in this domain, highlighting their characteristics and the corresponding up-to-date state-of-the-art results we have documented. (v) To provide further value, we included a range of datasets in the appendix that are pertinent to this domain.

---

### Official Review · Reviewer_NoL5 · 2023-07-27

**Soundness:** 3

**Excitement:**

2: Mediocre: This paper makes marginal contributions (vs non-contemporaneous work), so I would rather not see it in the conference.

**Paper Topic And Main Contributions:**

The paper presents an extensive review of recent and past literature in computer aided pronunciation training.   The paper is authoritative:  there is an overview of ' nuances of pronunciation' (section 2) with a collection and summary description of widely used data sets (Table 1).    The  modelling approaches in the 'Research Avenues' (section 4) are well chosen,  although the range and variety of material precludes  in-depth discussion.   When the paper does go into detail (e.g. description of Hierarchical Evaluation Structure in section 5),  the presentation is clear and informative.

The main contributions are claimed as:  revisit the literature following recent developments (line 98),  review qualitative studies that address intelligibility, comprehensiveness, and 'accentednes' (line 106), a review of available resources and evaluation measures (line 109), and suggestions for future work.

**Reasons To Accept:**

The paper assembles a large number of recent and not so recent papers into a systematic presentation that could be of use to speech processing experts who wish to review recent developments in automatic pronunciation assessment.   The literature covered is extensive and covers many aspects of the problem.

**Reasons To Reject:**

This may be due to length constraints, but  the paper requires extensive familiarity with recent work in acoustic modelling.    Techniques are mentioned without much explanation.    The target audience is not clear, but would seem to be experts in speech who wish to get up to date on automatic pronunciation assessment.   Other readers would have to do a good deal of background reading to follow this review.   As an example,  section 4.2 gives a one paragraph summary of work on 'Extended Recognition Networks (ERN)' without explaining what ERNs are.

The authors claim 'a discussion of remaining challenges and possible directions for future work' as one of their main contributions.   These are addressed only very briefly in section 6 (lines 594-604) as 1) lack of public resources and 2) unified evaluation metrics.    Neither point is profound or original.   The remainder of the section outlines possible applications that would become possible once computer-aided pronunciation training systems were viable.

Another motivation for the work is `With large transformer-based pre-trained models gaining popularity, re-visiting the existing literature and presenting a comprehensive study of the field is timely.'   However , apart from sections 4.5-4.8,  much of the paper is devoted to earlier work.  This isn't necessarily bad,  but the novelty relative to previous surveys isn't clear.

I have made some notes in the 'typos' sections below.  I would say that without extensive editing the paper is not presentable at EMNLP.



**Reproducibility:**

N/A: Doesn't apply, since the paper does not include empirical results.

**Reviewer Confidence:**

2: Willing to defend my evaluation, but it is fairly likely that I missed some details, didn't understand some central points, or can't be sure about the novelty of the work.

**Typos Grammar Style And Presentation Improvements:**

There are some unusual bibliography keys (e.g. 'cam', 'sin') and formatting errors (particularly casing) that should be fixed, as well as some incomplete entries (e.g. Caro 2023).

Grammatical error in abstract:  '...for both phonemic and prosodic.'

The title and/or abstract should make more clear that there is a focus on L2 pronunciation.

Table 1 includes SOTA results under a range of metrics for a collection of data sets.  These aren't really described in enough detail to be useful.   Some are defined in Section 5,  although superficially, e.g  at line 582  'Precision, Recall, and F-measure are widely used to evaluate the performance of MD models.'  and some are not explained at all (e.g. 'Min-Cost per phone').

Some acronyms (e.g. 'SSL') need definition.

---

> ### Author Rebuttal · Authors · 2023-08-29
>
> We would like to thank the reviewer for the feedback.
>
> ---------------------------------------------------------------------------------------------------------------------
> Q1: This may be due to length constraints, but the paper requires extensive familiarity with recent work in acoustic modelling. Techniques are mentioned without much explanation. The target audience is not clear, but would seem to be experts in speech who wish to get up to date on automatic pronunciation assessment. Other readers would have to do a good deal of background reading to follow this review. As an example, section 4.2 gives a one paragraph summary of work on 'Extended Recognition Networks (ERN)' without explaining what ERNs are.
>
> R1: The paper serves as a comprehensive introduction to the domain of automatic pronunciation assessment, and offers a valuable starting point for individuals entering this field with a background in speech. We meticulously outlined the diverse facets of pronunciation assessment, covering both phonetic and prosodic aspects of the field. We covered a comprehensive survey of old, revised, and current methodologies employed in modeling pronunciation assessment . Our objective is to introduce readers to a diverse range of existing methods, and resources in automatic pronunciation assessment.
>
> ---------------------------------------------------------------------------------------------------------------------
> ---------------------------------------------------------------------------------------------------------------------
> Q2: The authors claim 'a discussion of remaining challenges and possible directions for future work' as one of their main contributions. These are addressed only very briefly in section 6 (lines 594-604) as 1) lack of public resources and 2) unified evaluation metrics.
>
> R2: We would like to point it out that in addition to the challenges mentioned above (public resource and evaluation metrics), we also addressed the challenges in handling the following:
> Children's Speech (lines 636-642)
> The complexity introduced by the lack of a standard orthography for various languages/dialects (lines 646-648).
> Moreover, we emphasised the pivotal role of integrating Generative AI models to raise the reliability of CAPT tutoring (lines 608-618).
>
> We highlighted the significance of Multilingualism (lines 619-635), which facilitates not only the extension of knowledge across languages but also the transfer of insights to languages that are resource-scarce (lines 629-632).
>
> We acknowledge that we've provided an overview of existing challenges and a glimpse into future prospects. The essence of our paper centers on broad challenges rather than delving into specific, tailored ones.
>
> ---------------------------------------------------------------------------------------------------------------------
> ---------------------------------------------------------------------------------------------------------------------
> Q3: `With large transformer-based pre-trained models gaining popularity, re-visiting the existing literature and presenting a comprehensive study of the field is timely.' However , apart from sections 4.5-4.8, much of the paper is devoted to earlier work.
>
> R3: We would like to draw the reviewer's attention  regarding the content distribution within the paper. Sections of "previous approaches" (4.1-4.3) account for approximately half a page, the subsequent sections discussing "recent approaches" (4.4-4.8) are significantly more extensive, spanning around three pages, in which we offered a comprehensive depiction of the existing landscape concerning pronunciation assessment approaches in the current context.
>
> Furthermore, within the old/previous approaches, we discussed notable and recent advances in those framework/ techniques. For instance, in Section 4.4, specifically within (lines 315-323) and (lines 328-334), we presented strides in the realm of GOP reformulation.
>
> ---------------------------------------------------------------------------------------------------------------------
> ---------------------------------------------------------------------------------------------------------------------
>
> Q4: Table 1 includes SOTA results under a range of metrics for a collection of data sets. These aren't really described in enough detail to be useful. Some are defined in Section 5, although superficially, e.g at line 582 'Precision, Recall, and F-measure are widely used to evaluate the performance of MD models.' and some are not explained at all (e.g. 'Min-Cost per phone').
>
> R4:  We strongly believe that the information presented in Table 1 is a valuable representation of an extensive exploration across various benchmarks and datasets, summarising prevalent works. The table highlights how each dataset in the literature is  associated with distinct (sometimes unorthodox) evaluation metrics, which we have taken great care to meticulously outline in the 'SOTA' column along with concise results.
> It is important to acknowledge the plethora of evaluation metrics utilised in this field, many of which lack standardisation (e.g., Min-Cost per phone). For brevity, we were unable to provide an exhaustive elaboration on all these evaluation metrics and compensated this limitation by including references for further reading.
> We focused on the most widely used and recognised metrics such us (Phoneme Error Rate, F-measure, Precision, Recall, and Pearson Correlation Coefficient). These measures are becoming the de-facto metrics in the field over the last few years.
> We will give more details of these standard metrics in the  camera-ready version; we plan to enhance this section of evaluation metrics by incorporating comprehensive definitions and relevant equations to provide a more thorough understanding.

---

### Official Review · Reviewer_YAq7 · 2023-08-03

**Paper Topic And Main Contributions:** This is a survey paper on automatic p…
**Soundness:** 4

**Excitement:**

4: Strong: This paper deepens the understanding of some phenomenon or lowers the barriers to an existing research direction.

**Questions For The Authors:**

No questions on my side.

**Reasons To Accept:**

The survey is well-structured and clearly written.
Apart from being a survey it is also a nice introduction to automatic pronunciation assessment for someone new to the field.
Also, Figure 2 deserves a special credit.

**Reasons To Reject:**

I do not see any major reason to reject this paper.
I would prefer a longer section on evaluation metrics with some quantitative evidence on which metrics are prevalent in the surveyed papers, but the lack of such information does not disqualify this paper.

**Reproducibility:**

N/A: Doesn't apply, since the paper does not include empirical results.

**Reviewer Confidence:**

3: Pretty sure, but there's a chance I missed something. Although I have a good feel for this area in general, I did not carefully check the paper's details, e.g., the math, experimental design, or novelty.

---

> ### Author Rebuttal · Authors · 2023-08-29
>
> We would like to thank you for the encouraging feedback.
>
> ---------------------------------------------------------------------------------------------------------------------
> Q1: I would prefer a longer section on evaluation metrics with some quantitative evidence on which metrics are prevalent in the surveyed papers
>
> R1: We agree with your suggestion regarding a longer evaluation metrics section. Due to the limited page number in the initial submission, we couldn't provide more details on the reported evaluation metrics. We plan to address this concern in the camera-ready version of the paper. We will add more details on the hierarchical evaluation metrics, including detailed definition and equation (in main paper) and examples in Appendix.

---

### Meta-Review · Area_Chair_4s5T · 2023-09-19

**Recommendation:** 3

**Metareview:**

**Originality**

The presented paper is a very well written literature review of automatic pronunciation assessment. As such it is not original in and of itself.

**Significance**

The paper is a nice introduction to automatic pronunciation assessment for someone new to the field with some descriptive figures and tables thoroughly covering past approaches and describing current and future challenges. This is a useful resource though not something that has much immediate impact on the field.

**Clarity**

This paper is extremely thoughtful and well written.

**Pros:**
   - A very valuable introduction for those trying to enter the field of automated pronunciation assessment
   - Details current and past approaches as well as future challenges
   - The paper is incredibly well written.

**Cons:**
   - There are no new methods, data, etc., in the paper
   - Limited excitement

---

### Decision · Program_Chairs · 2023-10-07

**Decision:**

Accept-Findings

**Comment:**

**Originality**

The presented paper is a very well written literature review of automatic pronunciation assessment. As such it is not original in and of itself.

**Significance**

The paper is a nice introduction to automatic pronunciation assessment for someone new to the field with some descriptive figures and tables thoroughly covering past approaches and describing current and future challenges. This is a useful resource though not something that has much immediate impact on the field.

**Clarity**

This paper is extremely thoughtful and well written.

**Pros:**
   - A very valuable introduction for those trying to enter the field of automated pronunciation assessment
   - Details current and past approaches as well as future challenges
   - The paper is incredibly well written.

**Cons:**
   - There are no new methods, data, etc., in the paper
   - Limited excitement